# Increases in multiple resources promote competitive ability of naturalized non-native plants

Zhijie Zhang[1], Yanjie Liu [2][✉], Angelina Hardrath[1], Huifei Jin[2,3] & Mark van Kleunen [1,4]

Invasion by non-native plants is frequently attributed to increased resource availability. Still, our understanding is mainly based on effects of single resources and on plants grown without competition despite the fact that plants rely on multiple resources and usually grow in competition. How multiple resources affects competition between native and non-native plants remains largely unexplored. Here, with two similar common garden experiments, one in China and one in Germany, we tested whether nutrient and light availabilities affected the competitive outcomes, in terms of biomass production, between native and naturalized non-native plants. We found that under low resource availability or with addition of only one type of resource non-natives were not more competitive than natives. However, with a joint increase of nutrients and light intensity, non-natives were more competitive than natives. Our finding indicates that addition of multiple resources could greatly reduce the niche dimensionality (i.e. number of limiting factors), favoring dominance of non-native species. It also indicates that habitats experiencing multiple global changes might be more vulnerable to plant invasion.

[1] Ecology, Department of Biology, University of Konstanz, 78464 Konstanz, Germany. [2] Key Laboratory of Wetland Ecology and Environment, Northeast Institute of Geography and Agroecology, Chinese Academy of Sciences, 130102 Changchun, China. [3] University of Chinese Academy of Sciences, 100049 Beijing, China. [4] Zhejiang Provincial Key Laboratory of Plant Evolutionary Ecology and Conservation, Taizhou University, 318000 Taizhou, China. [✉]email: liuyanjie@iga.ac.cn

The rapid accumulation of naturalized non-native species is one of the characteristics of the Anthropocene[1,2]. Because some non-native species can threaten native species and disrupt ecosystem functioning[3], it has become an urgent quest to understand the mechanisms that allow non-natives to out-compete natives. One widely considered mechanism is the fluctuating resource hypothesis[4], which poses that 'a plant community becomes more susceptible to invasion whenever there is an increase in the amount of unused resources'. Although numerous studies have shown that resource increases can favor non-native plants over natives[5,6], most studies investigated the effect of a single resource, mainly nutrients, despite the fact that plants require different types of resources (e.g. nutrients, light). The studies that investigated the effects of multiple resources on native and non-native plants[7,8] usually looked at the growth of individually grown plants, instead of at competitive outcomes. Therefore, it remains largely unknown whether multiple resources interact in their effects on competitive outcomes between native and non-native plants.

How resources affect competition has long fascinated and puzzled ecologists[9,10]. Resource-competition theory predicts that if multiple species are competing for resources, the coexistence of all species is only possible when each species is limited by a different resource[11]. A classic example comes from algae, where *Asterionella formosa* and *Cyclotella meneghiniana* are able to coexist when *A. formosa* is limited by phosphate and *C. meneghiniana* is limited by silicate[12]. Resource addition (e.g. phosphate) will decrease the number of limiting resources and will thus favor the dominance of one species (known as the niche-dimension hypothesis *sensu* Harpole and Tilman 2007; Fig. 1). Although it remains challenging to identify limiting resources for more complex species (e.g. vascular plants), a few follow-up experiments have shown that coexistence of multiple plant species is less likely with the addition of multiple resources[13,14]. The explanation behind this is that the more types of resources are added, the more likely it is that the previously limiting resources are no longer limiting. The next step in this field of research is to predict which type of species will be favored with the addition of multiple resources.

One group of species that might benefit from the addition of multiple resources is naturalized non-native plants. First, most naturalized non-native plants originate from anthropogenic habitats[15], which are frequently rich in resources due to anthropogenic inputs (e.g. fertilizer spill-over) and due to disturbance. Consequently, successful non-natives are frequently those that are pre-adapted to high resource availability[16] and thus are favored by resource addition[6,17]. Second, non-natives might be limited by fewer factors than natives because their evolutionary history differs from natives[18–20]. For example, non-native plants might be released from and thus be less limited by natural enemies[21,22]. Such advantage of non-natives over natives may not be expressed when both types of species suffer from resource limitation, especially from the limitation of multiple resources[23]. However, the advantage will appear when resource limitation is removed by resource addition. Although resource-competition theory offers a potential mechanistic explanation of the success of non-native species, empirical tests remain rare.

Here, we conducted two experiments, one in Germany and one in China, with largely similar designs. In both locations, we grew multiple native and non-native native plant species (Supplementary Table 1) either alone, in monoculture, or in a mixture with one of the other species. To vary resource availabilities, we used two levels of fertilizer (i.e. nutrients) and two levels of light. We focused on nutrients and light for two main reasons. First, most successful non-native plants originate from, and also have naturalized in, nutrient-rich habitats[15,24], which indicates the importance of nutrients for plant invasion. Second, successful non-native plants are frequently larger than native plants[25], which might provide them with a superior ability to capture light and throw shade on neighboring native plants. Such an advantage might be more apparent in light-limited environments, where competition for light is more intense. We aimed to test whether resource availability affected pairwise competitive outcomes, in terms of biomass production, between native and naturalized non-native species. We found that under low resource availability or with the addition of only one type of resource non-natives were not more competitive than natives. However, with a joint increase in nutrients and light intensity, non-natives were more competitive than natives.

## Results

Overall, aboveground biomass production of plants significantly increased with nutrient availability (+66.6%; Fig. 2; Supplementary Table 2; $\chi^2 = 46.42$, $P < 0.001$), and marginally significantly increased with light intensity (+67.9%; $\chi^2 = 3.18$, $P = 0.075$). Moreover, aboveground biomass production increased the most with a joint increase of nutrient availability and light intensity (+79.4%), as indicated by the interaction between nutrient and light treatments ($\chi^2 = 24.12$, $P < 0.001$). Although averaged across competition treatments and the different light and nutrient treatments, non-native species tended to produce more aboveground biomass than native ones, this difference was not statistically significant ($\chi^2 = 2.04$, $P = 0.153$). This indicates that averaged across resource treatments, non-natives did not outcompete natives. However, the competitive outcome between natives and non-natives depended on the combination of nutrient and light treatments (Fig. 2; Supplementary Table 2; $\chi^2 = 4.66$, $P = 0.031$). More specifically, with a joint increase of nutrients and light intensity, non-natives produced 110.8% more aboveground biomass than natives; whereas this difference was much

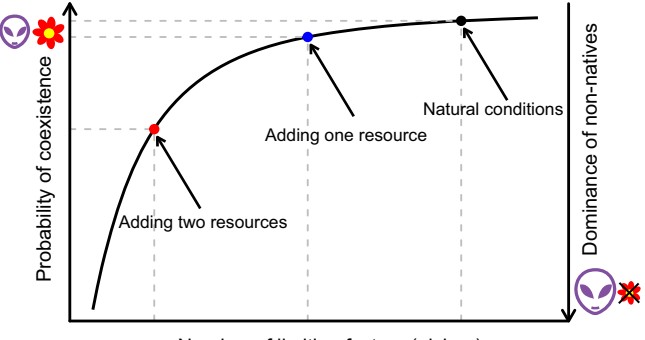

**Fig. 1 Graphical illustration of the potential effect of the number of limiting factors on species coexistence.** The probability of species coexistence increases with the number of limiting factors (according to the niche-dimension hypothesis), asymptotically approaching a 100% chance of coexistence. This is because the higher the number of limiting factors, the more likely that different species are constrained by different limiting factors. Under natural conditions (black dot), there are multiple limiting factors, such as nutrients, light, and enemies, favoring species coexistence. Adding one resource type will reduce the dimensionality of the niche space (i.e. number of limiting factors), thereby slightly reducing the probability of coexistence (blue dot). Adding multiple resource types will reduce the niche dimensionality, even more, greatly reducing the probability of coexistence and favoring the dominance of a single species (red dot). Because non-native species are likely to be pre-adapted to high resource availability and/or to be limited by fewer factors, their dominance is expected to be favored by the addition of multiple resource types.

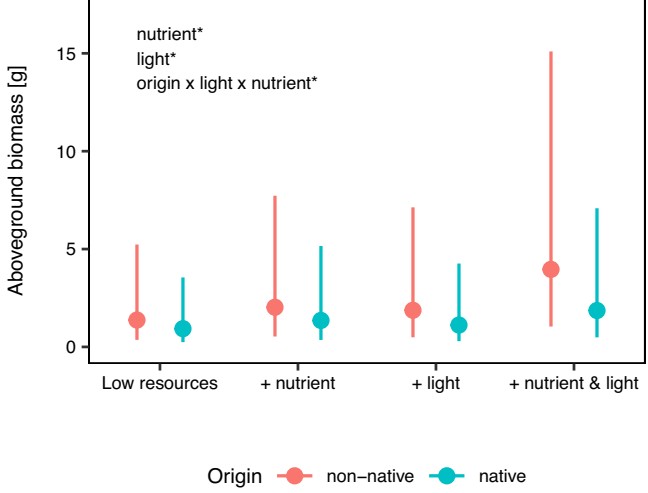

**Fig. 2 Effects of nutrient and light availabilities on competitive outcomes between native (blue) and non-native (red) plants.** The outcome is indicated by the difference in average biomass production between native and non-native plants across competition treatments (i.e. without competition, and with intra- and interspecific competition). For example, higher biomass production of non-native plants indicates that non-natives will outcompete natives. Error bars indicate 95% CIs. Significant effects are indicated in the left upper corner (see Supplementary Table 2 for details).

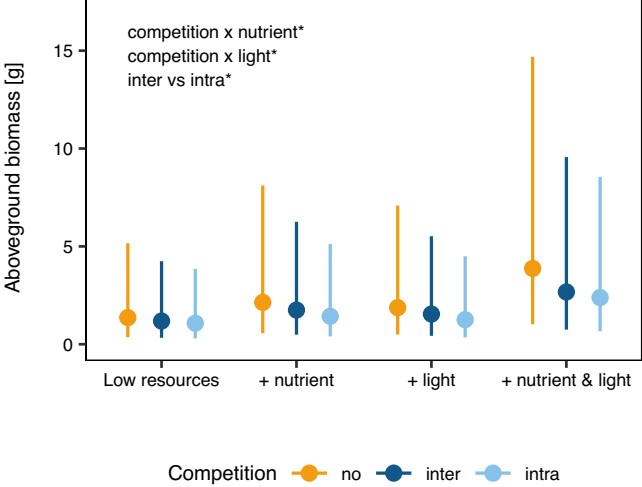

**Fig. 3 Effects of nutrient and light availabilities on effects of intra- and interspecific competition on plant aboveground biomass.** Yellow, dark blue, and light blue colors represent plants without competition, and with inter- and intraspecific competition, respectively. Error bars indicate 95% CIs. The effect of nutrient and light availabilities on intra- and interspecific competition did not significantly depend on the origin of the plants (see Supplementary Note 1 for details). Significant effects are indicated in the left upper corner (see Supplementary Table 2 for details).

smaller under low resource availability (+48.3%) or with the addition of only one type of resource (+48.4% under high nutrients only; +68.9% under high light intensity only).

Competition tended to reduce aboveground biomass production by 26.0%, as indicated by the difference between plants grown without competition and plants grown with competition (Fig. 3; Supplementary Table 2; $\chi^2 = 1.93$, $P = 0.165$). Although this effect of competition was not statistically significant averaged across the different nutrient or light treatments, it became more apparent with increased nutrients ($\chi^2 = 5.21$, $P = 0.022$) and

increased light intensity ($\chi^2 = 4.21$, $P = 0.040$). In addition, we found that plants produced 16.4% more aboveground biomass when competing with interspecific competitors than with intraspecific competitors (Fig. 3; Supplementary Table 2; $\chi^2 = 20.05$, $P < 0.001$).

## Discussion

Our experiments in China and Germany showed that under low resource availability, native and naturalized non-native plants did not significantly differ in aboveground biomass production across the competition treatments. This indicates that under those resource conditions non-natives will not outcompete the natives. Although an increase in one type of resource, either nutrients or light, increased biomass production, it affected natives and non-natives similarly, and thus did not change the potential competitive outcome. However, with a joint increase of nutrients and light intensity, non-natives produced more biomass than natives, indicating that non-natives may outcompete natives under the high availability of both resources. Our finding thus supports the fluctuating resource hypothesis, which posits that 'a plant community becomes more susceptible to invasion whenever there is an increase in the number of unused resources. Furthermore, our finding, along with those of others[17,26], explains why plant invasion is frequently associated with disturbance. This is because disturbance, by creating open patches, could result both in a flush of nutrients and a higher light intensity[27].

Our finding that across the two experiments, the addition of one type of resource did not favor non-native plants has several implications. First, it suggests that plants —irrespective of their origin— are limited by multiple factors, such as nutrients, light, water, and herbivory. In other words, niche space has multiple dimensions, each of which is represented by one limiting factor. While the addition of one resource type reduces the dimensionality of the niche space (e.g. adding nutrients can release plants from belowground resource competition, such as competition for nitrogen or phosphorus[13]), the remaining dimensions could still limit both native and non-native plants, allowing them to still coexist. Some previous studies, in line with our findings, showed that the addition of one type of resource (nutrients) did not favor non-native plants[28]. However, others found that the addition of nutrients only was sufficient to favor non-native plants[5,6]. One explanation for the apparent discrepancy could be that the former study was done in a greenhouse in winter, whereas the latter two studies were conducted under high-light conditions in summer. With the addition of nutrients, their environments were limited by neither nutrients nor light (same as the joint increases of nutrients and light in our study), which greatly reduced the dimensions of the niche space, favoring dominance by one of the two species.

The second implication of our finding is that native and non-native species did not strongly differ in their competitive abilities for nutrients and light. As the addition of one resource type reduces the dimensionality of the niche space, it intensifies competition for resources that form the remaining dimensions. For example, previous studies have shown that nutrient addition can intensify competition for light[29], which is asymmetric and therefore is more likely to result in competitive exclusion[30]. Indeed, we found that competition was more severe with nutrient addition (Fig. 3a). Consequently, if naturalized non-native plants have stronger abilities to compete for light than natives (e.g. when their tall stature allows them to intercept most light before it reaches the smaller natives[31]), they will dominate with nutrient addition. However, as this was not the case, we conclude that there was no strong difference in competitive abilities for light between the natives and non-natives in our experiments.

The two implications mentioned above raise the question of which factor or factors determine the higher competitiveness of naturalized non-native plants with joint increases of nutrients and light. One potential factor could be plant enemies, as we indeed did observe (although not measure) herbivory damage in our experiments. Because the evolutionary histories of non-native plants differ from those of native plants, non-native plants might be released from natural enemies[21]. This advantage might be stronger when other factors, for example, resource availability, are not limiting plant growth[23,32]. An alternative potential factor is the preadaptation of naturalized non-native plants. Many naturalized non-native plants occur in anthropogenic habitats[15], where resource availability is high due to human disturbance. Consequently, of the many non-native plants that have been introduced, the ones that managed to naturalize or become invasive are most likely the ones that were selected for high growth rates under high resource availability. Given that these two explanations are not mutually exclusive, future studies that test their relative importance, for example, by testing how non-natives from anthropogenic and those from natural habitats respond to plant enemies, are needed.

Naturalized non-native plants may not always exclude natives, even when light and nutrients are unlimited. This is indicated by the large variation in biomass differences between the competing native and non-native species in our experiments (e.g. in 38% of the species pairs, natives had higher biomass than non-natives when they competed with each other under high availability of light and nutrients), indicating that some of the natives can coexist with or even exclude some of the non-natives. Moreover, as we only measured short-term biomass differences instead of long-term changes in population sizes, which would also depend on survival and reproduction, we cannot say in how many cases the non-natives will ultimately exclude the natives. Still, biomass is usually positively related to reproductive output[33] and increases the chances of survival (more storage). Therefore, increases in biomass are likely to be indicative of higher fitness. Furthermore, the non-natives cannot increase infinitely as we found that intraspecific competition was stronger than interspecific competition. This implies that when a non-native species starts to dominate the community, it is likely to become self-limited until it becomes less dominant again. This calls for long-term studies, especially under natural conditions (e.g. by using locally collected plant material and by performing field experiments). Alternatively, one could do short-term studies that use a large range of densities of native and non-native plants (i.e. space-for-time-substitution), which mimics long-term community dynamics as well as the initial stage of invasion where non-native species start from low numbers.

The fluctuating resource hypothesis suggests that, with an increase in resources, a plant community becomes more susceptible to invasion. Our study suggests that this is particularly the case with increases of multiple resources, as this could greatly reduce the dimensionality of niche space, leading to competitive exclusion of one of the species. This can also explain why many studies have found that biological invasions are more frequent in disturbed, high-resource environments.

## Methods

**Study species**. To increase our ability to generalize the results, we conducted two multispecies experiments[34]. The experiments were designed independently, but, as they used similar treatments, we analyzed them jointly to further increase generalizability. For the experiment in China, we selected eight species that are either native or non-native in China (Supplementary Table 1). For the experiment in Germany, we selected 16 species that are either native or non-native in Germany (Supplementary Table 1). All 24 species, representing seven families, are herbaceous, mainly occur in grasslands, and are common in the respective regions. To control for phylogenetic non-independence of species, we selected at least one non-

native and one native species in each of the seven families. All non-native species are fully established (i.e. naturalized *sensu* Richardson et al.[35]) in the country where the respective experiment was conducted, and, as they are common, most of them could be considered invasive[36,37]. We classified the species as naturalized non-native or native to China or Germany based on the following databases: (1) "The Checklist of the Alien Invasive Plants in China"[38], (2) the Flora of China (www.efloras.org), and (3) BiolFlor (www.ufz.de/biolflor). Seeds or stem fragments of the study species were obtained from local botanical gardens, local commercial seed companies, or from wild populations (Supplementary Table 1).

**The experiment in China**. From 21 May to 27 June 2020, we planted or sowed the eight study species into plastic trays filled with potting soil (Pindstrup Plus, Pindstrup Mosebrug A/S, Denmark). We sowed the species at different times (Supplementary Table 1) because they were known to require different times until germination. Three species were grown from stem fragments because they mainly rely on clonal propagation, and the others were propagated from seeds (Supplementary Table 1).

On 13 July 2020, we transplanted the cuttings or seedlings into 2.5-L circular plastic pots filled with a mixture of sand and vermiculite (1:1 v/v). Three competition treatments were imposed: (1) competition-free, in which plants were grown alone; (2) intraspecific competition, in which two individuals of the same species were grown together; (3) interspecific competition, in which two individuals, each from a different species were grown together. We grew all eight species without competition, in intraspecific competition, and in all 28 possible pairs of interspecific competition. For the competition-free and intraspecific-competition treatments, we replicated each species seven times (i.e. we had seven technical replicates). For the interspecific-competition treatment, for which we had many pairs of species (i.e. biological replicates), we replicated each pair two times.

The experiment took place in a greenhouse at the Northeast Institute of Geography and Agroecology, Chinese Academy of Sciences (Changchun, China). The greenhouse had a transparent plastic film on the top, which reduced the ambient light intensity by 12%. It was open on the sides so that insects and other organisms could enter. To vary nutrient availability, we applied to each pot either 5 g (low-nutrient treatment) or 10 g (high-nutrient treatment) of a slow-release fertilizer (Osmocote® Exact Standard, Everris International B.V., Geldermalsen, The Netherlands; 15% N + 9% $P_2O_5$ + 12% $K_2O$ + 2% MgO + trace elements). To vary light availability, we used two cages (size: 9 m × 4.05 m × 1.8 m). One of them was covered with two layers of black netting material, which reduced the light intensity by 71% (low light-intensity treatment, where the light intensity was on average 233.5 µmol m$^{-2}$ s$^{-1}$, measured on a sunny day). The other was left uncovered (high light-intensity treatment, where the light intensity was on average 826.7 µmol m$^{-2}$ s$^{-1}$).

The experiment included a total of 672 pots ([8 no-competition × 7 replicates + 8 intraspecific-competition × 7 replicates + 28 interspecific-competition × 2 replicates]×2 nutrient treatments × 2 light treatments). The pots were randomly assigned to positions and were randomized once on 15 August within each block (i.e. the low or high light-intensity treatment). The initial height of each plant was measured on 15 July 2020, two days after the transplanting. We watered the plants daily to avoid water limitations. On 1 September 2020, we harvested the aboveground biomass of all plants. The biomass was dried at 65℃ for 72 h to constant weight and then weighed to the nearest mg.

**The experiment in Germany**. On 15 June 2020, we sowed seeds of the 16 species into plastic trays filled with potting soil (Topferde, Einheitserde Co). On 6 July 2020, we transplanted the seedlings into 1.5-L pots filled with a mixture of potting soil and sand (1:1 v/v). Like the experiment in China, we imposed three competition treatments: competition-free, intraspecific competition, and interspecific competition. However, in this experiment, which had two times more species than the experiment in China, we only included 24 randomly chosen species pairs for the interspecific-competition treatment, and all of these pairs consisted of one naturalized non-native and one native species. For the competition-free treatment, we replicated each species two times (i.e. we had two technical replicates). For the competition treatments, we did not use technical replicates for any of the species combinations for logistic reasons. However, as we had a large number of species pairs in the inter-specific competition treatment, we had many biological replicates.

The experiment took place outdoors in the Botanical Garden of the University of Konstanz (Konstanz, Germany). To vary nutrient availability, we applied to each pot once a week either 100 ml of a low-concentration liquid fertilizer (low-nutrient treatment; 0.5‰ Universol ® Blue oxide fertilizer, 18% N + 11% P + 18% K + 2.5% MgO + trace elements) or 100 ml of a high-concentration of the same liquid fertilizer (high-nutrient treatment; 1‰). In total, pots in the low- and high-nutrient treatment received 0.4 and 0.8 g fertilizer, respectively. To vary light availability, we used eight metal wire cages (size: 2 m × 2 m × 2 m). Four of the cages were covered with one layer of white and one layer of green netting material, which reduced the ambient light intensity by 84% (low light-intensity treatment; where the light intensity was on average 219.0 µmol m$^{-2}$ s$^{-1}$, measured on a sunny day). The remaining four cages were covered only with one layer of the white netting material, which served as a positive control for the effect of netting and reduced light intensity by 53% (high light-intensity treatment; where the light intensity was on average 678.4 µmol m$^{-2}$ s$^{-1}$). In other words, the low light-intensity treatment received 34% (66% reduction) of the light intensity in the high light-intensity treatment.

The experiment included a total of 320 pots ([16 no-competition × 2 replicates + 16 intraspecific-competition + 32 interspecific-competition]×2 nutrient treatments × 2 light treatments). The eight cages were randomly assigned to fixed positions in the botanical garden. The pots were randomly assigned to the eight cages (40 pots in each cage) and were re-randomized once within and across cages of the same light treatment on 3 August 2020. Besides the weekly fertilization, we watered the plants two or three times a week to avoid water limitations. On 7 and 8 September 2020, we harvested the aboveground biomass of all plants. The biomass was dried at 70 °C for 96 h to constant weight and then weighed to the nearest 0.1 mg.

**Statistical analyses.** All analyses were performed using R version 3.6.1[39]. To test whether resource availability affected competitive outcomes between native and non-native species, we applied linear mixed-effects models to analyze the biomass of the plants in the two experiments jointly and separately, using the *nlme* package[40]. For the model used to analyze the two experiments jointly, we excluded interspecific competition between two non-natives and interspecific competition between two natives from the experiment in China, because non-native-non-native and native-non-native combinations were not included in the experiment in Germany. When we analyzed each experiment separately, the results were overall similar to the results of the joint analysis. Therefore, we focus in the manuscript on the joint analysis and present the results of the separate analyses in Supplementary Note 2.

Because plant mortality was low and mainly happened after transplanting, we excluded pots in which plants had died. The final dataset contained 1180 individuals from 871 pots. In the model, we included the aboveground biomass of individuals as the response variable. We included the origin of the species (non-native or native), competition treatment (see below for details), nutrient treatment, light treatment and their interactions as fixed effects; study site (China or Germany), and identity and family of the species as random effects. In addition, we allowed each species to respond differently to the nutrient and light treatments (i.e. we included random slopes). To account for pseudoreplication[41], we also included pots as random effects and cages (ten cages, eight from Germany and two from China) as random block effects. In the competition treatment, we had three levels: (1) no competition, (2) intraspecific competition, and (3) interspecific competition between native and non-native species. To split them into two contrasts, we created two variables[42] testings (1) the effect of the presence of competitors, and (2) the difference between intra- and interspecific competition (see Supplementary Note 3 for details). To improve the normality of the residuals, we natural-log-transformed aboveground biomass. To improve the homoscedasticity of the residuals, we allowed the species and competition treatment to have different variances by using the *varComb* and *varIdent* functions[43]. Significances of the fixed effects were assessed with likelihood-ratio tests (type II) with the *car* package[44].

To determine the 'competitive outcome', i.e. which species will exclude or dominate over the other species at the endpoint for the community[45,46], one should ideally conduct a long-term study. Alternatively, one could vary the density of each species, which mimics the dynamics of species populations across time (see refs. [47,48] for examples). However, applying this space-for-time-substitution method would have largely increased the size of the experiment, especially when combined with the light and nutrient treatments. Still, by growing plants alone, in intraspecific competition and in interspecific competition, our experiments meet the minimal requirement for measuring competitive outcome, at least in terms of short-term biomass production[46,49].

In the linear mixed-effects model of individual biomass, a significant effect of origin would indicate that native and naturalized non-native species differed in their biomass production, across all competition and resource-availability (light and nutrients) treatments. This would tell us the competitive outcome between non-natives and natives across different resource availabilities. For example, an overall higher level of biomass production of non-native species would indicate that non-natives would dominate when competing with natives. A significant interaction between a resource-availability treatment and the origin of the species would indicate that resource availability affects the biomass production of native and non-native species differently, averaged across all competition treatments. In other words, it would indicate that resource availability affects the competitive outcome between natives and non-natives. A significant interaction between a resource-availability treatment and the competition treatment would indicate that resource availabilities modify the effect of competition (e.g. no competition vs. competition). Other studies frequently have inferred competitive outcomes from the effect of competition by calculating the relative interaction intensity[50]. However, while the competitive outcome and effect of competition are often related, they are not equivalent[45]. This is because the competitive outcome is both determined by the effect of competition and intrinsic growth rate[48,49]. For example, a plant species that strongly suppress other species but has a low intrinsic growth rate still cannot dominate the community.

**Reporting summary**. Further information on research design is available in the Nature Research Reporting Summary linked to this article.

## Data availability
The data in the study are archived in Figshare (https://doi.org/10.6084/m9.figshare.21269589)[51].

## Code availability
The code in the study is archived in Figshare (https://doi.org/10.6084/m9.figshare.21269589)[51].

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

## Acknowledgements

We thank O. Ficht, M. Fuchs, X. Zhang, L. Wang, and Y. Li for their practical assistance. Z.Z. acknowledges funding from the China Scholarship Council (201606100049) and support from the International Max Planck Research School for Organismal Biology. Y.L. acknowledges funding from the Chinese Academy of Sciences (Y9B7041001).

## Author contributions

Z.Z. and Y.L. conceived the idea. Z.Z., Y.L., and M.v.K. designed the study. A.H. and H.J. performed the experiments. Z.Z. led the analyses and writing, with inputs from Y.L. and M.v.K.

## Competing interests

The authors declare no competing interests.
