## [Peer Review File · Communications Biology]

Reviewers' comments:

Reviewer #1 (Remarks to the Author):

I reviewed the manuscript for the second time. I still think the paper by Zhang and colleagues "Increases in multiple resources promote plant invasions" addresses the important and timely question whether invasive species vs. native species differ in their response to the addition of multiple resources and how this is translated into differences in competitive ability. I think the topic is of wider interest as it may explain why alien species become invasive in certain contexts or under certain environmental conditions. I also see that by manipulating multiple resources (not a single one as done already) that may be associated with biological invasions (i.e. light, nutrients) this study delivers novel knowledge for understanding biological invasions. The results are presented clearly and the statistical approach is valid overall.

Generally, the revision of the manuscript is carefully done and I have only minor comments.

I like the conceptual figure that is now included in the manuscript. Please note that there is a mistake in the axis (i.e. "of" is double)

As the experiment is short-term and only covers two levels of frequency (i.e. monocultures and mixtures), I would be careful to use the term "space-for-time" substitution in this context (see response of the authors to one of my comments). The experiment only consists of two levels of frequency (i.e. 50% (mix), 100% (mono)) and is clearly missing a low frequency treatment (lower than 50%). It therefore does not allow species to increase when rare (invasion criterion, see e.g. Crawley, M. J. (Ed.). (2009). Plant ecology). However, I appreciate that the authors state that the "experiments meet the minimal requirement for measuring competitive outcome".

Reviewer #2 (Remarks to the Author):

Zhang et al. examines the role of soil nutrients and light on the competitive outcome of non-native and native plants (24 total) in Germany and China. Their analyses, which combined both experiments (which were similar but not exact), showed that it took both soil nutrients and light for the non-native plants to consistently outcompete the native species. Their experiment is interesting and the use of two sites (Germany and China) is compelling (though they were greenhouse studies). However, the authors frame the paper as one that examines two types of resources, but in truth one treatment added many different soil nutrients (each a different resource) and the other treatment added light. This needs to be clarified, and because of this, they misuse the niche dimensionality hypothesis in the discussion. Also, it isn't clear in the paper if the focus of their framework should be alien (or better termed as non-native) or invasive plant species. These issues and the following comments need to be addressed.

Specific Comments/Edits:

1. Title: Since this research didn't look at invasion dynamics explicitly, authors should not say "...promote plant invasion". Maybe "plant invasion potential" or "competitive ability".
2. L25 and elsewhere: 'availabilities' should be 'availability' in ALL cases
3. L42: Consider changing alien to exotic or non-native as alien is considered a loaded term by some. Usually better to go with non-native.
4. L45: delete "proposed by"

5. L51: delete comma

6. L50-52: restate this sentence because there a lot of studies that look at the effect of multiple resources on invasive plant success. I think the authors want to focus on the lack of studies of multiple resources on competitive outcomes between exotic and native species.

7. L72: delete "the one of"

8. L 73-76: Confusing statement – consider revising. You state that aliens have an advantage over natives when both suffer from resource limitation, but then state the advantage of the alien will appear when resource limitation is removed.

9. L82 'First' is spelled wrong

10. L84-86: OK, but why manipulate light since invaders will modify the light environment – it is part of their competitive advantage. Defense of the light treatment should be better explained

11. L95: Here is where terminology can become an issue. Alien (or non-native) doesn't mean invasive which, by the title, this manuscript is focusing on. This should be explained in the intro or references to alien should be changed to invasive species. Similarly, L104-105: How was this determined? Better to use regional or local databases/websites to identify which species were actually invasive.

12. L97: Might be useful to also include functional group status in Table S1

13. L115: Were some of the plants already 6 weeks old at the time of the start of the experiment?

14. L129-132: The nutrient additions were surprising for 2 reasons:

1) The authors didn't have a control with no nutrients, as 5 and 10 g isn't that dissimilar. Please justify.

2) From the way the paper was introduced, it seemed they were going to use multiple soil nutrients separately, instead of a mix (As L57-66 seemed to be setting up – since types (L64) referred to different soil nutrients). It seems like the authors try to present the design as 2 types of resources, when it really is just two treatments since one has many resources added (the nutrient treatment) and the other with just one (the light treatment). The authors need to clarify their framework throughout the manuscript and be consistent.

15. L157-159: Fertilization should be reported in the same units across the two experiments.

16. L165: 53% full sun is NOT 66% less light than 84% full sun. This is easier to see with specific numbers (rather than percentages of percentages). If full sun is 2000 $\mu\text{Mol m}^{-2} \text{s}^{-1}$, then the high light treatment will be at 1680 and the low light will be at 1060 $\mu\text{Mol m}^{-2} \text{s}^{-1}$. This is a 37% reduction, not 66%.

17. L176-L185: The steps the authors took (removing the native-native and invasive-invasive comparisons in China, and comparing the joint analysis to the separate analyses) make me reasonably certain that their findings would be robust to the differences in experimental design between the two locations. However, they may get better results, and it may be more appropriate, if they account for the variation in treatments. Instead of both experiments having a categorical high/low light treatment (since these were not equivalent), both could use a quantitative measure (0%, 71%, 53%, 84% reductions). And same thing for the nutrients (if treatment level differed between sites) - instead of categorical use a numeric variable.

18. L193-L194: Including cages and pots as random effects addresses pseudoreplication as the

authors state, but I am not sure how this works with the China experiment. As I understand the methods, there was only one high-light cage and one low-light cage in China. If a random effect is included both for the cage effect and for the location (China), then including a fixed effect for light environment will result in a non-identifiable (overfit) model. I think the authors need to clarify how this analysis was done because based on my understanding, it is not possible.

19. L196-198: Please provide explanation as to how you calculated your dummy variables, (which could just be changed to "variables")

20. L201: Likelihood ratio tests in the car package in R allows specifying which type sums of squares (I, II, or III) are to be used. Which type sums of squares used is information that should be included in these methods, as that affects interpretation.

21. L256: This statement somewhat ignores allelopathic effects (if this is referring to when plants were grown alone).

22. L257: Technically, this was an increase in many types of resources (nutrients) and another type of resource (light)

23. L271: Again, each type of soil nutrient (N, P, K, etc.) is one dimension (Harpole et al. 2016), so it is not what the experiment did. Again, in L282-283. Please revise.

24. L294: Was herbivory damage quantified? If so, please include.

25. L308: How many times or what percent of species combinations resulted in natives excluding invasives? Please include.

Reference

Harpole, W. S., Sullivan, L. L., Lind, E. M., Firn, J., Adler, P. B., Borer, E. T., ... & Wragg, P. D. (2016). Addition of multiple limiting resources reduces grassland diversity. *Nature*, 537(7618), 93-96.

Dear Editor,

We are pleased that you gave us the opportunity to revise our manuscript, and that the reviewers found our work interesting, compelling and important. We used their helpful comments to improve our manuscript. Below, we give point-by-point responses to the comments of the reviewers.

Our responses are in blue. All changes in the manuscript text file are highlighted.

Sincerely,

Zhijie Zhang (on behalf of all authors)

Reviewers' comments:

Reviewer #1 (Remarks to the Author):

I reviewed the manuscript for the second time. I still think the paper by Zhang and colleagues “Increases in multiple resources promote plant invasions” addresses the important and timely question whether invasive species vs. native species differ in their response to the addition of multiple resources and how this is translated into differences in competitive ability. I think the topic is of wider interest as it may explain why alien species become invasive in certain contexts or under certain environmental conditions. I also see that by manipulating multiple resources (not a single one as done already) that may be associated with biological invasions (i.e. light, nutrients) this study delivers novel knowledge for understanding biological invasions. The results are presented clearly and the statistical approach is valid overall.

Generally, the revision of the manuscript is carefully done and I have only minor comments.

Response: We thank the reviewer for the positive evaluation and the compliments.

I like the conceptual figure that is now included in the manuscript. Please note that there is a mistake in the axis (i.e. “of” is double)

Response: We have now corrected the typo.

As the experiment is short-term and only covers two levels of frequency (i.e. monocultures and mixtures), I would be careful to use the term „space-for-time” substitution in this context (see response of the authors to one of my comments). The experiment only consists two levels of frequency (i.e. 50% (mix), 100% (mono)) and is clearly missing a low frequency treatment (lower than 50%). It therefore does not allow species to increase when rare (invasion criterion, see e.g. Crawley, M. J. (Ed.). (2009). Plant ecology). However, I appreciate that the authors state that the “experiments meet the minimal requirement for measuring competitive outcome”.

Response: We have now mentioned the low frequency treatment in the discussion (L336).

Reviewer #2 (Remarks to the Author):

Zhang et al. examines the role of soil nutrients and light on the competitive outcome of non-native and native plants (24 total) in Germany and China. Their analyses, which combined both experiments (which were similar but not exact), showed that it took both soil nutrients and light for the non-native plants to consistently outcompete the native species. Their experiment is interesting and the use of two sites (Germany and China) is compelling (though they were greenhouse studies). However, the authors frame the paper as one that examines two types of resources, but in truth one treatment added many different soil nutrients (each a different resource) and the other treatment added light. This needs to be clarified, and because of this, they misuse the niche dimensionality hypothesis in the discussion. Also, it isn't clear in the paper if the focus of their framework should be alien (or better termed as non-native) or invasive plant species. These issues and the following comments need to be addressed.

Response: We agree that adding fertilizer may remove multiple resource dimensions (e.g. nitrogen and phosphorus). However, even if that is the case, the fertilizer treatment in combination with the light treatment still removed a larger number of dimensions than either the fertilizer or the light treatment. Consequently, we believe that our concept still holds. We have now revised the discussion and legend of Fig. 1 to make this clearer (L295-297 & 485-488).

Our paper focuses on naturalized alien species. We have now changed the title to avoid confusion.

We are aware that some people prefer the term ‘non-native’ over ‘alien’, but as ‘alien’ is the term used in the unified framework for biological invasions proposed by Blackburn et al. (2011; Trends in Ecology and Evolution 26:333-339), we prefer to stick to ‘alien’. However, if the editor insists, we are willing to replace it with ‘non-native’.

Specific Comments/Edits:

1. Title: Since this research didn’t look at invasion dynamics explicitly, authors should not say “...promote plant invasion”. Maybe “plant invasion potential” or “competitive ability”.

Response: we have now used ‘competitive ability’.

2. L25 and elsewhere: ‘availabilities’ should be ‘availability’ in ALL cases

Done.

3. L42: Consider changing alien to exotic or non-native as alien is considered a loaded term by some. Usually better to go with non-native.

Response: We are aware that some people prefer the term ‘non-native’ or ‘exotic’ over ‘alien’, but as ‘alien’ is the term used in the unified framework for biological invasions proposed by Blackburn et al. (2011; Trends in Ecology and Evolution 26:333-339), we prefer to stick to ‘alien’. However, if the editor insists, we are willing to replace it with ‘non-native’.

4. L45: delete “proposed by”

Done.

5. L51: delete comma

Done.

6. L50-52: restate this sentence because there a lot of studies that look at the effect of multiple resources on invasive plant success. I think the authors want to focus on the lack of studies of multiple resources on competitive outcomes between exotic and native species.

Done (L49-51).

7. L72: delete “the one of”

Done.

8. L 73-76: Confusing statement – consider revising. You state that aliens have an advantage over natives when both suffer from resource limitation, but then state the advantage of the alien will appear when resource limitation is removed.

Response: We meant that aliens do not have an advantage over natives when both suffer from resource limitation. We have revised it (L75).

9. L82 ‘First’ is spelled wrong

Corrected.

10. L84-86: OK, but why manipulate light since invaders will modify the light environment – it is part of their competitive advantage. Defense of the light treatment should be better explained

Response: We now explain that competition for light will be more serious in light-limited environments (L88-89).

11. L95: Here is where terminology can become an issue. Alien (or non-native) doesn't mean invasive which, by the title, this manuscript is focusing on. This should be explained in the intro or references to alien should be changed to invasive species. Similarly, L104-105: How was this determined? Better to use regional or local databases/websites to identify which species were actually invasive.

Response: There are multiple definitions of “invasive”, which as main criteria use either the impact of the alien species on natives or their spread rate. In practice, both are frequently unknown. Because a widespread alien that is local abundant is likely to have impact and have spread rapidly (Catford et al. 2016, van Kleunen et al. 2018), the alien species used in our study could be considered invasive. Still, since different databases/websites use different definitions (e.g. according to neobiota.bfn.de, among the eight alien study species in Germany, only *Epilobium ciliatum* is classified as invasive in Germany, although most of the other ones are also widely naturalized), we try to avoid using ‘invasive’. We now emphasize more strongly that all alien species are naturalized, and we have also changed the title accordingly.

12. L97: Might be useful to also include functional group status in Table S1

Done.

13. L115: Were some of the plants already 6 weeks old at the time of the start of the experiment?

Response: We cannot exclude this. The longest period between sowing and transplanting was 8.5 weeks. This was for the two *Solidago* species. These species were sown earlier than the other species because they usually need a long time to germinate. So, their plants most likely were younger than 8.5 weeks old at transplanting. The information about the sowing dates of each species can be found in Table S1.

14. L129-132: The nutrient additions were surprising for 2 reasons:

1) The authors didn't have a control with no nutrients, as 5 and 10 g isn't that dissimilar. Please justify.

2) From the way the paper was introduced, it seemed they were going to use multiple soil nutrients separately, instead of a mix (As L57-66 seemed to be setting up – since types (L64)

referred to different soil nutrients). It seems like the authors try to present the design as 2 types of resources, when it really is just two treatments since one has many resources added (the nutrient treatment) and the other with just one (the light treatment). The authors need to clarify their framework throughout the manuscript and be consistent.

Response: We did not consider a no-nutrient control because the plants would not grow at all (the substrate was a sand-vermiculite mixture, which hardly contains any nutrients). As the plant biomass was higher in the 10-g-fertilizer treatment than the 5-g- fertilizer treatment, we believe that the fertilizer treatments were appropriate.

Regarding the comment on number of added resources, please, see our response to the same comment of this reviewer above.

15. L157-159: Fertilization should be reported in the same units across the two experiments.

Done (L166).

16. L165: 53% full sun is NOT 66% less light than 84% full sun. This is easier to see with specific numbers (rather than percentages of percentages). If full sun is $2000 \mu\text{Mol m}^{-2} \text{ s}^{-1}$, then the high light treatment will be at 1680 and the low light will be at $1060 \mu\text{Mol m}^{-2} \text{ s}^{-1}$. This is a 37% reduction, not 66%.

Response: The low and high light-intensity treatments received 47% and 16% of full sun light (-53% vs -84%), respectively. So, the low light-intensity treatment received 34% (-66%) of the light intensity in the high light-intensity treatment. We have rephrased this section to avoid confusion (L170 & 173-175).

17. L176-L185: The steps the authors took (removing the native-native and invasive-invasive comparisons in China, and comparing the joint analysis to the separate analyses) make me reasonably certain that their findings would be robust to the differences in experimental design between the two locations. However, they may get better results, and it may be more appropriate, if they account for the variation in treatments. Instead of both experiments having a categorical high/low light treatment (since these were not equivalent), both could use a quantitative measure

(0%, 71%, 53%, 84% reductions). And same thing for the nutrients (if treatment level differed between sites) - instead of categorical use a numeric variable.

Response: While we agree it will be interesting to use light and nutrients as quantitative measures, we are afraid that we cannot do this for the following reasons:

As we have clarified above, the light conditions did not differ strongly between the two experiments. More specifically, measured on a sunny day, the low light-intensity treatment received on average 233 and 219 $\mu\text{mol}/\text{m}^2/\text{s}$ in China and Germany, respectively. The high light-intensity treatments received on average 827 and 678 $\mu\text{mol}/\text{m}^2/\text{s}$ in China and Germany, respectively. We have now added this information to the manuscript (L137-139 & 169-175).

Regarding the nutrient treatment, different fertilizers were used. We used a slow-release fertilizer in China, and a liquid fertilizer in Germany (this is because the experiments were designed independently, as we have mentioned in the MS). Although the amount of applied fertilizer differed between the two experiments, the amounts of released nutrients, which was not measured, may not differ in the same way. Consequently, we are not able to assign a quantitative measure to the experiment in China.

Importantly, however, even if we would have quantitative measures, they would be fully confounded with the random term study site. In other words, the random term study site already accounts for differences in the light and fertilizer treatments between the two experiments.

18. L193-L194: Including cages and pots as random effects addresses pseudoreplication as the authors state, but I am not sure how this works with the China experiment. As I understand the methods, there was only one high-light cage and one low-light cage in China. If a random effect is included both for the cage effect and for the location (China), then including a fixed effect for light environment will result in a non-identifiable (overfit) model. I think the authors need to clarify how this analysis was done because based on my understanding, it is not possible.

Response: When analyzing the two experiment together, there are ten cages (eight from Germany and two from China), which are nested within location. Because each light treatment consists of five cages, this does not cause any problem in the model.

However, when analyzing the experiment in China separately, the cage effect could not be included because it is identical to the light treatment. We have now explained this in the supplement (Supplement S2 L37-38)

19. L196-198: Please provide explanation as to how you calculated your dummy variables, (which could just be changed to “variables”)

Response: We have now provided an explanation in the Supplement S3. We deleted ‘dummy’.

20. L201: Likelihood ratio tests in the car package in R allows specifying which type sums of squares (I, II, or III) are to be used. Which type sums of squares used is information that should be included in these methods, as that affects interpretation.

Response: It is Type II, and we now mention this (L212).

21. L256: This statement somewhat ignores allelopathic effects (if this is referring to when plants were grown alone).

Response: This referred to when plants compete. We have now rephrased it to clarify this (L268).

22. L257: Technically, this was an increase in many types of resources (nutrients) and another type of resource (light)

Response: Please, see our response to the same comment above.

23. L271: Again, each type of soil nutrient (N, P, K, etc.) is one dimension (Harpole et al. 2016), so it is not what the experiment did. Again, in L282-283. Please revise.

Response: Please, see our response to the same comment above.

24. L294: Was herbivory damage quantified? If so, please include.

Response: Herbivory damage was not quantified. It was based on observation. We now mention this explicitly (L308).

25. L308: How many times or what percent of species combinations resulted in natives excluding invasives? Please include.

Response: Because our experiment is not long-term, we can not predict competitive exclusion. However, we calculated the percent of species pairs where natives had higher biomass than aliens (i.e. natives were more competitive; L320-323).

Reference

Harpole, W. S., Sullivan, L. L., Lind, E. M., Firn, J., Adler, P. B., Borer, E. T., ... & Wragg, P. D. (2016). Addition of multiple limiting resources reduces grassland diversity. *Nature*, 537(7618), 93-96.

REVIEWERS' COMMENTS:

Reviewer #1 (Remarks to the Author):

The authors did a great job with their revision and I don't have additional comments.

Reviewer #2 (Remarks to the Author):

This revised manuscript is much improved and the authors have satisfactorily responded to all comments. However, there are some minor edits needed. The edits referring to resource type is to keep their reference to their treatments consistent and to increase clarity.

L49 add "the" before effect.

L283 add "type" after resource

L295 add "type" after resource

L485 add "type" after resource

L497 change to "multiple resource types"

Dear editor,

We are delighted that you conditionally accepted our manuscript. We have taken your suggestion, changing ‘alien’ into ‘non-native’. Below, we give point-by-point responses to the comments of the reviewers. Our responses are in blue.

Sincerely,

Zhijie Zhang (on behalf of all authors)

REVIEWERS' COMMENTS:

Reviewer #1 (Remarks to the Author):

The authors did a great job with their revision and I don't have additional comments.

Response: We thank the reviewer for the compliment.

Reviewer #2 (Remarks to the Author):

This revised manuscript is much improved and the authors have satisfactorily responded to all comments. However, there are some minor edits needed. The edits referring to resource type is to keep their reference to their treatments consistent and to increase clarity.

Response: We thank the reviewer for her/his suggestions.

L49 add “the” before effect.

Done.

L283 add “type” after resource

Done.

L295 add “type” after resource

Done.

L485 add “type” after resource

Done.

L497 change to “multiple resource types”

Done.